# Dual-Functional Liposomes with Carbonic Anhydrase IX Antibody and BR2 Peptide Modification Effectively Improve Intracellular Delivery of Cantharidin to Treat Orthotopic Hepatocellular Carcinoma Mice

**DOI:** 10.3390/molecules24183332

**Published:** 2019-09-12

**Authors:** Xue Zhang, Congcong Lin, Waikei Chan, Kanglun Liu, Aiping Lu, Ge Lin, Rong Hu, Hongcan Shi, Hongqi Zhang, Zhijun Yang

**Affiliations:** 1Institute of Translational Medicine, Medical College, Yangzhou University, Yangzhou 225001, China; zhangxueflora@163.com (X.Z.); prhurong@163.com (R.H.); shihongcan@yzu.edu.cn (H.S.); 2School of Chinese Medicine, Hong Kong Baptist University, Hong Kong, China; 14485680@life.hkbu.edu.hk (C.L.); nickie@hkbu.edu.hk (W.C.); liukanglun@hkbu.edu.hk (K.L.); aipinglu@hkbu.edu.hk (A.L.); 3The Key Laboratory of Syndrome Differentiation and Treatment of Gastric Cancer of the State Administration of Traditional Chinese Medicine, Yangzhou 225001, China; 4Department of Medicinal Chemistry and Natural Medicine Chemistry, College of Pharmacy, Harbin Medical University, Harbin 150081, China; 5Changshu Research Institute, Hong Kong Baptist University, Changshu Economic and Technological Development (CETD) Zone, Changshu 215505, China; 6School of Biomedical Sciences, Chinese University of Hong Kong, Hong Kong, China; linge@cuhk.edu.hk

**Keywords:** hepatocellular carcinoma, dual-functionalized liposomes, carbonic anhydrase IX, BR2 peptide, cantharidin

## Abstract

Liposomal nanotechnology has a great potential to overcome the current major problems of chemotherapy. However, the lack of penetrability and targetability retards the successful delivery of liposomal carriers. Previously, we showed that BR2 peptide modification endowed cantharidin-loaded liposomes with intracellular penetration that enhanced the drug cytotoxic effects. Here, we aimed to improve the targeting delivery of drugs into cancer cells via highly expressed carbonic anhydrase IX (CA IX) receptors by modifying our previous catharidin-loaded BR2-liposomes with anti-CA IX antibody. A higher cellular uptake of dual-functional liposomes (DF-Lp) than other treatments was observed. Induction of CA IX over-expressing resulted in a higher cellular binding of DF-Lp; subsequently, blocking with excess antibodies resulted in a decreased cancer-cell association, indicating a specific targeting property of our liposomes towards CA IX expressed cells. After 3h tracking, most of the liposomes were located around the nucleus which confirmed the involvement of targeting intracellular delivery. Cantharidin loaded DF-Lp exhibited enhanced cytotoxicity in vitro and was most effective in controlling tumor growth in vivo in an orthotopic hepatocellular carcinoma model compared to other groups. Collectively, our results presented the advantage of the BR2 peptide and CA IX antibody combination to elevate the therapeutic potential of cantharidin loaded DF-liposomes.

## 1. Introduction

Hepatocellular carcinoma (HCC) is still a major health problem worldwide with morbidity and mortality rates increasing [1]. Though different therapeutic modalities have been tried, the survival outcomes of patients are still disappointing, especially when the disease is at an advanced stage [2]. Systemic therapy is regarded as the only option for these cases and sorafenib is the only drug approved as the first-line chemotherapeutic drug for systemic treatment of HCC. However, conventional chemotherapeutic agents have not exhibited any meaningful therapeutic benefit, particularly in overall survival and the quality of life [3,4,5]. The reasons for these poor outcomes are mainly contributed by the non-specific accumulation of drugs in the tumor, high toxicity to normal tissues, and acquired drug resistance of the chemotherapeutics [6]. 

Liposomal drug delivery systems (DDS) have recently been explored for cancer treatment to overcome the limitations related to systemic applications of therapeutic agents [7]. The rationale for the targeted delivery system approach to treat HCC has been proposed for years and the evidences to support its use in cancer treatments include the pivotal role that the nanocarriers have in the delivery process, such as the improvement of drug solubility, the alteration of pharmacokinetic distribution, particular accumulation into the tumor sites by enhanced permeability, and retention (EPR) effects [4,8,9]. 

Extensive research has been performed with immunoliposome targeting systems for cancer treatment in vitro and in vivo, demonstrating that immunoliposomes binding selectively to antigen-specific target cells to deliver therapeutic compounds to the cells [10]. Nevertheless, new challenges have occurred from the design of these immunoliposomes, such as liposome internalization into the tumor cells, restricted diffusion, and penetration through the tumor tissue, non-specific binding by serum proteins, and translation from pre-clinical animal models to clinical studies [11]. Moreover, the identified ligands for anticancer treatment do not embrace all cancer types to date, HCC in particular. 

Carbonic anhydrase IX (CA IX), a cell surface enzyme, has been found over-expressed in many solid tumors vis-à-vis their corresponding normal tissues [11], including hepatocellular carcinoma [11,12,13]. This kind of tumor-specific protein has attracted much attention from researchers as a potential candidate for enzyme-targeted anticancer drug delivery [14]. BR2 is a derivative of the nonspecific cell-penetrating anticancer peptide buforin IIb, which showed higher efficiency of penetration into cancer cells than into normal cells compared with well-known penetrating peptide Tat [15]. Cantharidin, a sesquiterpene derivative, is isolated from several species of beetles, primarily the dried body of Chinese medicine blister beetles (*Mylabris phalerata* or *M. cichorii*), the cardinal beetles, along with the Spanish fly *Lytta vesicatoria* [16]. In China, CTD as the main constituent of mylabris has been extensively used for treatment of hepatoma and oesophageal carcinoma for more than 2000 years [17,18]. Previously, we reported that BR2-modified liposomes improved intracellular penetrability of CTD-liposomes to the cancer cells [15]. The intracellular delivery was however limited with this only one specific ligand modification, perhaps due to the fact that the specific ligand-mediated endocytosis pathway was often saturated [19]. A targeted liposomes delivery system with anti-CA IX antibody and CPP33 to deliver triptolide for non-small cell lung cancer is successfully relevant reinforcing the use of dual functionalities [14]. Considering the limited reports on dual-targeted systems with both anti-CA IX antibody and BR2 in HCC treatment, we hypothesized that the surface modification of cantharidin-loaded liposomes with BR2 and CA IX antibody (DF-Lp), will improve the selectivity of the liposomes toward the over-expressed CA IX and help better cytosolic cantharidin delivery leading to enhanced anti-cancer effects, both, in vitro and in vivo. In addition, this study made one step further to carry out the experiments in orthotopic HCC HepG2 tumor model instead of subcutaneous HCC xenograft, as in our previous study. 

## 2. Results and Discussion

### 2.1. Preparation and Characterization of Dual-Functionalized Liposomes (DF-Lp)

The dual-functional liposomal delivery system (DF-Lp) in our present study was developed with the post-insertion approach (Figure 1) based on the translocation of DSPE-PEG-CA IX micellar lipids in exchange for liposomal bilayers [20]. Firstly, the activated BR2 peptide was coupled to the DSPE-PEG-Mal lipid and the successful synthesis was verified with a right-shifted peak appeared in the mass spectra by MALDI-TOF-MS analysis (Appendix A). Subsequently, the BR2-modified liposomes were prepared by ethanol injection method for the following conjugation. The intact anti-CA IX Ab was then chemically reduced with DTT solution to expose the thiol group for maleimide group reaction. The successful reduction was confirmed as showing a half molecular weight of 75 kD when compared to the whole antibody with a molecular weight of approximately 150 kD in Coomassie blue staining SDS-PAGE (Figure 1). 

The reduced CA IX Ab was immediately incubated with DSPE-PEG-Mal micelle overnight at 4 °C to form the anti-CA IX Ab conjugated DSPE-PEG lipid. The resultant was then confirmed with the upper-shifted molecular weight compared to the half-antibody on SDS-PAGE (Figure 1). Finally, the CTD loaded dual-functional liposomes (DF-Lp/CTD) was obtained by conducting the post-insertion method in which the anti-CA IX Ab conjugated micelle were incubated with the pre-formed CTD loaded BR2-liposomes at 60 °C for 2 h. In this way, the Ab ligand would be presented at the outer surface of the liposomes and maintain its binding capacity [21,22]. Afterward, free anti-CA IX Ab and micelle were removed by Sepharose CL-4B column. 

The finished liposomes appeared to be homogenous suspensions with good dispersion and with a sustained release profile (Appendix A). As shown in Table 1, the average particle size of DF-Lp/CTD was 98.3 ± 1.8 nm. This particle size result indicated that the ligands had been conjugated to the liposomal surface as there was a slight increase in particle size when compared to the single-ligand modified liposomes. The DF-Lp/CTD with such a nano-size could facilitate the passive accumulation due to the EPR effects of rapidly grown tumor vasculature [23]. The percentage of CTD encapsulated in liposomes was around 80%.

### 2.2. Cell Association of NBD-DPPE-Labeled Dual-Functional Liposomes

The cell association of the DF-Lp delivery system with HepG2 cells were imaged by CLSM. Liposomes were labeled with NBD-DPPE fluorescent lipid. By comparison to the three different formulations, more liposomes (green fluorescence) were clustered around the cellular nuclei (blue fluorescence) (Figure 2), and the translocation into cellular cytoplasm for DF-Lp was prominently higher than the rest of the groups (Appendix A) (*p* < 0.05), suggesting a higher drug delivery ability of our DF-Lp delivery system. 

### 2.3. Analysis of CA IX Antibody Directed Specific Cellular Uptake of Dual-Functional Liposomes

To investigate the specific cellular uptake of DF-Lp, we incubated the HepG2 cells under different oxygen supply condition to induce CA IX expression. CA IX receptors were expressed at high levels in hypoxic HepG2 cells using western blot but much lower in normoxic condition (Figure 3A). Next, we used these two cell lines to assess the specific binding properties of DF-Lp since anti-CA IX Ab was incorporated for the specific affinity. We incubated the NBD-DPPE-labeled DF-Lp for 3 h and evaluated the binding efficacy by CLSM. As shown in Figure 3B, the fluorescence within the CA IX positively expressed HepG2 cells was in a large amount than those CA IX negative HepG2 cells (*p* < 0.05) (Appendix A). It should be mentioned that although CA IX is negatively expressed in normoxia condition, the DF-Lp with both BR2 peptide and anti-CA IX antibody modification could still bind to HepG2 cells due to its lipid composition and BR2 ligand [15]. 

### 2.4. Analysis of CA IX Receptor-Mediated Endocytosis of Dual Functional Liposomes

To further address the specificity of DF-Lp for CA IX receptors, we performed a competition assay on CA IX positively expressed HepG2 cells. As shown in Figure 3C, when an excess amount of anti-CA IX Ab was added to compete with DF-Lp, the intracellular fluorescence within CA IX positively expressed cells was decreased. The decrease of fluorescence intensity reached significance when compared to the group without anti-CA IX antibody addition (Appendix A). Moreover, with unspecific antibody pre-treated, the intracellular fluorescence was obviously high compared to that with anti-CA IX antibody pre-treated groups. These results demonstrated an excellent cell-specific efficacy of DF-Lp and also suggested that the anti-CA IX Ab, with specific binding to CA IX receptors on the HCC cells, is probably effectively driving the endocytosis of DF-Lp. 

### 2.5. Cellular Internalization of Dual Functional Liposomes by Cancer Cells

As it was reported that endocytosis mediated by receptors usually led to endosome formation [24], and the drug is possible to be deactivated in the endosomes per se, it is essential to avoid DDS degradation in endosomes in order to complete its delivery to kill the cancer cells [25]. In our study, the colocalization of DF-Lp with endosomes/lysosomes was detected with endosomes/lysosomes stained by LysoTracker Red DND-99 to estimate the endosome escaping ability of the liposome. As shown in Figure 4, green fluorescence represented NBD-DPPE labeled DF-Lp carriers, and the red fluorescence represented the endosomes/lysosomes. For initial 0.5 h incubation of the DF-Lp with HepG2 cells, only a few DF-Lp entered the cells whereas the majority remained in endosomes/lysosomes (points by white arrow). After incubation for 3 h, in contrast, only a small portion of liposomes were still trapped in endosomes/lysosomes, and large area of green fluorescence was seen diffusing in the cytoplasm, demonstrating that DF-Lp had the ability to reach its ultimate target cytoplasm, which is in line with a previous report that BR2 liposome could successfully bypass endosomes/lysosomes and transfer into the cytosol [26]. As such, our study indicated that DF-Lp could be a good carrier for therapeutic agent delivery to the cell cytoplasm and purposefully kill the cancer cells.

Anti-proliferative effects of the tested liposomes in vitro were evaluated on HepG2-red-Fluc cells using bioluminescence and MTT assay. As shown in Figure 5, the inhibition properties of DF-Lp/CTD showed both in the bioluminescence and MTT results in a dose-dependent change on the HepG2-red-Fluc cells. Besides, DF-Lp/CTD showed more significant inhibition on cell viability than the free CTD group (*p* < 0.01), which might be due to the enhanced solubility of the drug after liposomal encapsulation. Moreover, compared to the BR2-Lp/CTD and CA IX-Lp/CTD, DF-Lp/CTD at low concentration showed more cytotoxicity to HepG2-red-Fluc cells (*p* < 0.05), which should be attributable to the improved targeting efficacy of the bi-ligand modified liposomes. These results indicated the potent effects of this DF-Lp/CTD in vitro. 

### 2.6. Orthotopic HCC Model for In Vivo Testing

The orthotopic HCC with luciferase-expressing cells provides a great model for basic and clinical research [27]. It allows the detection of tumor growth in the body by the IVIS Lumina system. 

In this study, the luciferase activities of HepG2-red-Fluc cells were examined in vitro first. As shown in Figure 6A,B, the luciferase expression in HepG2-red-Fluc was clearly seen in a cell number dependent manner. Their bioluminescence detected by the imaging system was positively correlated well with increasing cell numbers. The larger the cell number, the higher the bioluminescence, suggesting that it is appropriate to use the bioluminescence intensity in vivo to indicate the tumor size.

Having demonstrated that in vitro delivery of the dual-functional liposomes (DF-Lp) could produce pronounced effects, we further evaluate the efficacy of these liposomes induced anticancer potential in vivo. After the intrahepatic injection of the cells, the successful inoculation of cells could be seen as a clear bleb on the liver surface (Appendix A). The successful establishment of orthotopic HCC model was further demonstrated by IVIS Lumina Imaging System after intraperitoneal injection of D-luciferin (150 mg/kg), which was taken to capture the visible light photograph and luminescent image (Appendix A). The liver specimen of the same mouse was also examined (Appendix A). These results indicated that this orthotopic HCC mice model was feasible to be used in subsequent experiments.

### 2.7. In Vivo Tumor Distribution of DF-Lp

The fluorescent dye DiR was used to label the DF-Lp, and long-circulation and specific accumulation of the DF-Lp was evaluated by observing the red fluorescence in mice and the increasing fluorescence intensity at the liver part with IVIS imaging system at 4 h and 24 h post-injection. The fluorescence images in Figure 6C revealed the distribution of DiR-labeled DF-Lp while bioluminescent images show the tumor location. The fluorescence from DiR-labeled DF-Lp was detectable in the liver at 4 h after injection. An increase of fluorescence signals in the liver part was also observed, indicating an accumulation of liposomes. The fluorescence signals also remained visible after 24 h injection, demonstrating the probably prolonged retention of liposomes in tumors. These findings revealed that our liposomal delivery system can achieve long-circulating time and targeted delivery to HCC tissue by intravenous administration, indicating that our dual-functionalized liposomal system is a suitable nanocarrier for chemotherapeutic agents delivery.

### 2.8. Tumor Growth Inhibition Study of DF-Lp/CTD

To evaluate whether the DF-Lp/CTD could deliver CTD to the tumor site to inhibit HCC tumor growth, we used an orthotopic HCC mice model that expresses luciferase. Different CTD-formulations were administrated intravenously for four doses. The tumor size was shown as the luciferase expression detected by the IVIS Imaging system at a six-day interval, as shown in Figure 7A. In Figure 7B, the representative images show that while the bioluminescence intensity was continuously increased at the tumor sites in the saline control group, it was obviously lower in the DF-Lp/CTD treated groups. To further quantify bioluminescence in all groups, the total flux value of each mice in each group (*n* = 5) was drawn against the time (Figure 7C), where the saline-treated mice had a sharply elevated bioluminescence, while the free CTD and liposomal CTD systems treated mice overall showed much lower bioluminescence indicating their therapeutic effects. In Figure 7C, it could be seen that the curve of DF-Lp/CTD group exhibited a slow rise compared to other groups, although the difference was not significant. This group had obvious and consistent low bioluminescence than all other groups starting from day 12, reaching ~8 fold and significant reduction vis-à-vis saline and free CTD treated mice at day 24 (*p* = 0.046) (Figure 7C). Although the bioluminescence in BR2-Lp/CTD treated group went sharply down from day 18, it did not reach a significance when compared to saline-treated group or other groups. 

In comparison, mono-functionalized liposomal CTD had a somewhat lower (CA IX-Lp/CTD) and slower (BR2-Lp/CTD) reduction of bioluminescence and the reduction was not significantly different when compared to the control group, indicating that dual-functionalized liposomal CTD is most powerful for the treatment of HCC. In contrast, mono-functionalized liposomal BR2-Lp/CTD could achieve similar effects later than DF-Lp/CTD, reflecting its lack of power active targeting, whereas the CA IX-Lp/CTD effect is much weaker, reflecting its lack of power of penetration into cancer cells to kill. Hence, in line with what we proposed, DF-Lp/CTD appears to be a good DDS for the better treatment of HCC. 

In term of the toxicity of the different formulations in this study, the bodyweights of all drug-treated groups were comparable during the treatment and there were no obvious differences among the groups, implying that all the CTD-loaded liposomes had no differences in systemic toxicity (Figure 7D).

## 3. Materials and Methods

### 3.1. Materials

Soybean lecithin (SPC) was purchased from Shanghai Tai Wei Chemical Company (Shanghai, China). DSPE-PEG_2000_ was bought from Avanti Polar Lipids (Alabaster, AL, USA). DSPE-PEG-Mal (SUNBRIGHTDSPE-0.20MA) and NBD-DPPE (COATSOME FE-6060NB) was purchased from NOF Co. Ltd. (Tokyo, Japan). BR2 peptide, cys-RAGLQFPVGRLLRRLLR, was provided by SciLight Biotechnology (Beijing, China). Cantharidin (CTD) was obtained from Chengdu Biopurify Phytochemicals Ltd. (Sichuan, China). The rabbit monoclonal anti-CA IX antibody targeting the N-terminal region of the protein that is exposed to the extracellular side was obtained from Abcam (Cambridge, UK). The goat anti-rabbit horseradish peroxidase (HRP)-conjugated secondary antibody was obtained from Bio-rad Laboratories (Hercules, CA, USA). D-luciferin was purchased from Onwon Inc., Hong Kong, China. Hoechst 33342 and the fluorescent LysoTracker Red DND-99 was purchased from Molecular Probes Inc. (Eugene, OR, USA). The lipophilic near-infrared fluorescent dye 1,1′-dioctadecyltetramethyl indotricarbocyanine iodide (DiR) used for labeling liposomes was supplied by Caliper LifeSciences (Hopkinton, MA, USA). All other reagents are purchased from Sigma otherwise it was specified.

### 3.2. Cell Lines

The human HCC cell line HepG2 cells were cultured in Dulbecco’s modified Eagle’s medium (DMEM) supplemented with 10% fetal bovine serum (FBS) and 100 IU/mL penicillin, and 100 mg/mL streptomycin (Life Technologies, Carlsbad, CA, USA), at 37 °C incubator. To track the transplanted cells in vivo, HepG2-red-Fluc cells were purchased from PerkinElmer (Waltham, MA, USA) and the cells were transfected with firefly luciferase gene from *Luciola italica* (Red-FLuc), and maintained in Eagle’s MEM (ATCC) containing 10% FBS at 37 °C in the incubator.

### 3.3. Animals

Male BALB/c athymic (nu/nu) nude mice (5–7 weeks old; weight, 17–20 g) were bought from Tin Hang Technology Limited (Hong Kong, China) and housed under pathogen-free conditions, fed standard food, and given free access to sterilized water. Mice were acclimatized for 7 days after arrival. All experimental procedures were done according to guidelines of the Committee on the Use of Human & Animal Subjects in Teaching & Research of Hong Kong Baptist University and the Health Department of the Hong Kong Special Administrative Region. The ethical approval for the project is FRG2/14-15/082.

### 3.4. Preparation of Single-Ligand and Dual-Ligand Modified Liposomes

#### 3.4.1. Synthesis of DSPE-PEG-BR2 Conjugates

BR2 peptide was conjugated to the distal end of DSPE-PEG-Mal group according to the previous method [15]. Briefly, BR2 peptide with a cysteine (14.5 mg) was dissolved in 4 mL of HEPES buffer (20 mM HEPES, 10 mM EDTA-2Na, pH 6.5). Then, 20 mg of DSPE-PEG-Mal was dissolved in absolute ethanol and dried by rotary evaporation in 60 °C water bath to form the dried lipid film. Then the film was hydrated, and the micelle was reacted with BR2 peptide at a ratio of 1:1 (mol/mol). After agitation for 48 h, the resulting solution was dialyzed, lyophilized, and stored for the following experiment.

Successful formation of DSPE-PEG_2000_-BR2 was identified using a matrix-assisted laser desorption/ionization-time of flight (MALDI-TOF) mass spectrometer (Autoflex III; Bruker Daltonics Inc., Billerica, MA, USA).

#### 3.4.2. Synthesis of DSPE-PEG-CA IX Conjugated Micelle

Firstly, the activated half anti-CA IX Antibody (Ab) was produced by dithiothreitol (DTT) for conjugation to the DSPE-PEG-Mal group using the sulfhydryl reactive chemistry at room temperature for 90 min [28]. Then, 5 times volume ethyl acetate was added to the mixture to extract DTT 3 times. The DSPE-PEG-Mal micelles were prepared by a dried lipid film-hydration method by the hydration of DSPE-PEG-Mal in PBS (pH 6.6, 0.01M EDTA) with heating at 65 °C for 30 min. The reduced antibody was conjugated to DSPE-PEG-Mal micelles by incubation at 4 °C with gentle agitation overnight. The successful reduction of the whole anti-CA IX Ab and the conjugation of DSPE-PEG-Mal-CA IX were mixed with sample loading buffer, loaded onto a non-reducing sodium dodecyl sulfate-polyacrylamide gel electrophoresis (SDS-PAGE), as confirmed by the Coomassie blue staining.

#### 3.4.3. Preparation and Characterization of Liposomal CTD

CTD-loaded dual-functional liposomes were prepared with the post-insertion method. The CTD-loaded BR2-liposomes (BR2-Lp/CTD) were firstly prepared by the ethanol injection method according to our previous study [15]. Briefly, lipids of SPC/DSPE-PEG_2000_/DSPE-PEG_2000_-BR2 (molar ratio of 96:2:2) with or without CTD (at a lipid-to-drug ratio of 80:1) were dissolved in ethanol and subsequently hydrated with 5 mL of PBS (pH 7.4) by rapid injection. After stirring for 1 h for liposome formation, the suspensions were extruded repeatedly through an extruder first with a 200 nm-sized polycarbonate filter and then a 100 nm-sized filter. Then the antibody ligands were incorporated into the preformed liposomes, with or without BR2-modification, by the post insertion approach (Figure 1) to obtain anti-CA IX-Ab modified liposomes (CA IX-Lp/CTD) and dual-functionalized liposomes (DF-Lp/CTD). Briefly, pre-formed liposomes and DSPE-PEG_2000_-CA IX micelles were co-incubated for 2 h at 60 °C. Free Ab or micelles were then removed by gel filtration on a Sepharose CL-4B column (Sepharose CL-4B, Sigma-Aldrich; Empty PD-10 Column, GE Healthcare) [28]. To provide control values, the same thermal treatments were applied to other liposome suspensions, such as without bi-functional ligands or only with BR2-peptide modification.

To confirm the conjugation of the anti-CA IX Ab to liposomes surface, SDS-PAGE electrophoresis was performed and the samples with sample buffer were then loaded onto each lane of the 10% SDS-PAGE. Staining was performed with Coomassie Brilliant Blue R-250 Dye for 1h followed by washing in distilled water for 2 h. 

Besides, in the formulations, based on different experimental purposes, the different fluorescent liposomes were prepared with NBD-DPPE-fluorescent lipid or DiR fluorescent dye labeling. NBD-DPPE-labeled liposomes for in vitro studies were prepared by substitution of partial lipids by NBD-DPPE lipid [28]. The DiR-labeled liposomes were prepared as above, with the fluorescent dye being dissolved in ethanol for an in vivo targeting experiment. Then, the liposomes prepared in our study were characterized by particle size. The particle size and polydispersity index (PDI) of different liposomes were characterized by a Delsa Nano HC Particle Analyzer (Beckman Coulter, Brea, CA, USA). The CTD concentration in liposomes were analyzed by GC–MS and the encapsulation efficiency was calculated according to the previous study [15]. 

### 3.5. In Vitro Studies

#### 3.5.1. Induction of CA IX Expression In Vitro

HepG2 cells (1 × 10^6^) were incubated either under standard culture condition (5% CO_2_ at 37 °C for normoxia) or under hypoxic condition (constituting O_2_, CO_2_, and N_2_ with 1%, 5%, and 94%, respectively, at 37 °C for hypoxia induction) with a chamber. The CA IX expressed in hypoxia or normoxia incubated HepG2 cells were then detected by using western blot. 

#### 3.5.2. Cell Association of NBD-DPPE Labeled Dual-Functional Liposomes

Cellular association of DF-Lp with HepG2 cells were analyzed and compared to BR2-Lp, CA IX-Lp by a confocal microscopy. Briefly, HepG2 cells were cultured on glass-bottom dishes for total adhesion. Then, fresh medium containing different NBD-DPPE labeled liposomes were added and cells were incubated at 37 °C for another 3 h. Then, the cells were washed with PBS thrice, fixed with 4% paraformaldehyde (PFA) for an additional 20 min and further stained with Hoechst 33342 (5 μg/mL) for another 10 min. Finally, the cells were imaged by a Leica TCS SP8 confocal laser scanning microscopy (CLSM). The cellular uptake of liposomes were quantified by ImageJ software with Fiji package (NIH, version 1.51 d, NIH, Bethesda, MD, USA).

#### 3.5.3. Analysis of CA IX Antibody Directed Specific Cellular Uptake of Dual Functional Liposomes

To evaluate the CA IX Ab-directed specific cellular uptake, HepG2 cells with CA IX-positive and CA IX-negative expression was used. NBD-DPPE labeled DF-Lp were incubated with these different expressed CA IX cells. Briefly, 5 × 10^5^ HepG2 cells/well were seeded into glass-bottom dishes and incubated under hypoxic and normoxic conditions to generate CA IX-positive and CA IX-negative cells, respectively. Then cells were washed with PBS and exposed to dual-functionalized NBD-DPPE-labeled dual-functional liposomes for 3 h. Subsequently, cells were washed and fixed using 4% PFA for 20 min, stained with Hoechst 33342 for another 10 min followed by washing before to fluorescent microscopy. The fluorescence intensity was quantified by using Image-J software with Fiji package (NIH, version 1.51 d). 

#### 3.5.4. Analysis of CA IX Receptor-Mediated Endocytosis of Dual-Functional Liposomes

HepG2 cells (~100,000) were seeded in six-well plates and incubated under hypoxia overnight in complete medium for competing assay. Cells were then incubated in serum-free medium for 30 min, blocked with 5% bovine serum albumin (BSA) diluted with serum-free medium for 30 min at 37 °C, then pre-incubated with an excess of anti-CA IX primary Ab (1 mg/mL, 1:50) for 15 min in serum-free medium. After washing with PBS, cells were incubated with NBD-DPPE labeled DF-Lp for 30 min at 37 °C, then fixed for 20 min in 4% cold PFA and stained with Hoechst 33342 (5 μg/mL) for cell nuclei staining for another 10 min. Liposome binding was visualized by a fluorescence microscopy. 

#### 3.5.5. Cellular Internalization of Dual-Functional Liposomes by Cancer Cells

To track the internalization and endosomal release of liposomes, cells were treated with NBD-DPPE labeled DF-Lp for 0.5 h and 3 h. Then, the old medium was replaced with 100 nM LysoTracker Red in complete medium for an additional 0.5 h incubation at 37 °C to stain endosome/lysosome. With the same protocol mentioned above, the cells were washed, fixed, stained and imaged using the CLSM.

### 3.6. In Vitro Cytotoxicity Study

An in vitro cytotoxicity assay was conducted by bioluminescence monitoring and MTT assay. Briefly, 8 × 10^3^ HepG2-red-Fluc cells/well were dispensed into a black 96-well plate (Corning, Corning, NY, USA) in triplicates in five groups (control, free CTD, BR2-Lp/CTD, CA IX-Lp/CTD, DF-Lp/CTD). After incubation for 24 h, 50 μL of 150 μg/mL d-luciferin substrate was added prior to measurement. The bioluminescence light output was captured using the IVIS Spectrum system (Caliper, Hopking, MA, USA). The exposure time was set to auto and readings were quantified using the Living Image v4.4 software. The light output was represented in terms of average radiance (photons/seconds/cm2/steradian). The lower value indicated fewer live cells. For MTT-based cell viability assay, HepG2-red-Fluc cells were seeded in a transparent 96-well plate with 8 × 10^3^ cells/well. After allowing 24 h incubation for cell attachment, different preparation solutions were diluted appropriately in fresh medium and added to cells for co-incubation for 24 h, then 20 μL of 5 mg/mL MTT was added for additional 4 h incubation, followed by adding 100 μL of DMSO and subjected to the microplate reader at 570 nm. 

### 3.7. In Vivo Performance of Therapeutic Efficacy of DF-Lp

#### 3.7.1. Othotopic Tumor Model in Mice

In order to establish an orthtopic HCC model, the luciferase-transfected HepG2-red-Fluc cells were obtained and the luciferase activities were determined. After that, the orthotopic HepG2 tumor model was established by direct intrahepatic injection of HepG2-red-Fluc cells according to the literature with a slight modification [29]. Briefly, mice were anesthetized, positioned in supine on the procedure table, then, a small transverse incision below the sternum was made, and a portion of the liver was exposed. HepG2-red-Fluc cells were suspended in DPBS containing 50% Matrigel at about 1 × 10^6^ cells in 40 μL per mouse and slowly injected into the hepatic lobe using 1-mL syringe with 29-gauge needle. After injection, the abdominal incision was then closed with a 5-0 suture [30]. 

#### 3.7.2. Biodistribution Studies

The in vivo biodistribution of DF-Lp was conducted with a fluorescent dye DiR labeling. After checking the tumor in the nude mice by IVIS imaging with d-luciferin i.p. injection, a single dose of DF-Lp/DiR was i.v. injected via the tail vein at DiR dose of 2 mg/kg. Then the fluorescent images of mice were taken with the IVIS spectrum system at 4 h and 24 h after DF-Lp/DiR administration.

#### 3.7.3. Tumor Growth Inhibition Studies

HepG2-red-Fluc cells bearing nude mice were allocated into five treatment groups (*n* = 5 mice in each group) after tumor cell inoculation for 7 days (Figure 6A). Mice in different groups were treated with the following preparations, respectively, saline, CTD solution, BR2-Lp/CTD, CA IX-Lp/CTD, and DF-Lp/CTD. All the preparations were administrated via the tail vein injection every 3 days for a total of 4 doses, and the tumors were monitored by IVIS imaging systems every 6 days with D-luciferin (150 mg/kg) i.p. injection. The CTD at a dose of 0.4 mg/kg was administrated. The body weights of mice were recorded every 6 days as a systemic drug toxicity indicator.

### 3.8. Statistical Analysis

The experimental data were presented as the mean ± standard deviation (SD) for in vitro and mean ± standard error of mean (SEM) for in vivo data. The significance of the data was analyzed using GraphPad Prism 6.01 with ANOVA. The differences were considered statistically significant when the *p*-value was less than 0.05.

## 4. General Conclusions

The overall purpose of this study was aimed at the development of a drug delivery platform with improved intracellular delivery and enhanced cancer cell-specificity. Our earlier studies showed that surface modification of cantharidin liposomes with cell-penetrating peptide BR2, CPP33, helped liposomes penetrate into tumor cells [14,15]. The intracellular delivery was however limited with this only one ligand modification and the high specific targeting for cancer cells is also desired [15,19]. To address this challenge, we designed dual-functional liposomes, by adding a second ligand, anti-CA IX antibody, selective for CA IX receptors, known to be over-expressed in the cell-surfaces of most in vivo tumor sites.

Dual-functional liposomes targeting HCC were successfully constructed by incorporation of BR2 and anti-CA IX Ab onto PEGylated liposomes surface. To validate the CA IX Ab-directed specific targeting of DF-Lp to tumor cells, the NBD-DPPE-labeled liposomes were used initially. In vitro cellular binding of the NBD-DPPE labeled liposomes suggested that the DF-Lp had a higher cellular affinity than the other single ligand modified liposomes. In comparison to the CA IX-positive with negative expressed HepG2 cells, we have revealed that the DF-Lp had a specific binding property to CA IX positive cells. A low cellular binding when competing with excess antibodies indicated that the specific binding of the DF-Lp might come from the CA IX receptor-mediated intracellular endocytosis. Besides, from the co-localization results of both liposomes and the endosomes/lysosomes, the DF-Lp appeared to be able to avoid the degradation of endosomes/lysosomes, showing a good potential for it to be a drug carrier. Through in vitro cell viability study, we have demonstrated that this newly designed cantharidin loaded dual-functional liposomes had a higher cell growth inhibition on HepG2 cells than the other tested formulations. In addition, the long-time circulation in vivo of the DF-Lp could be seen in the biodistribution imaging, which again provided strong evidence of the promise of this dual-functional liposome in delivering HCC treatment. More importantly, the in vivo anticancer efficacy of this cantharidin loaded liposome in orthotopic HCC models showed an enhanced tumor growth inhibition compared to other treatments, with the dual-functional liposomal cantharidin being the best among the three liposomal formulations. These results are encouraging to show that this dual-functional liposome is a potent liposomal delivery system for targeted drug delivery, justifying further development for its clinical use on HCC patients after proper clinical trials and critical evaluation.

## Figures and Tables

**Figure 1 molecules-24-03332-f001:**
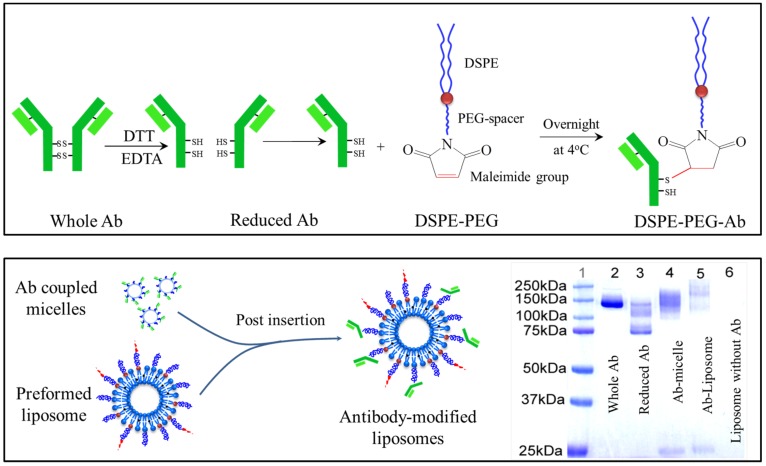
Schematic illustrations of DSPE-PEG-anti-CA IX Ab conjugation and the verification of the reduced antibody conjugation to micelles or liposomes by SDS-PAGE analysis with Coomassie blue staining.

**Figure 2 molecules-24-03332-f002:**
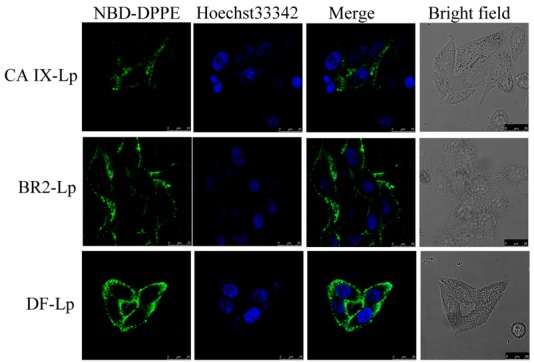
Cell association of NBD-DPPE labeled DF-liposomes compared to BR2-Lp and CA IX-Lp were imaged by CLSM after 3 h incubation with HepG2 cells. Scale bar = 25 μm. NBD-DPPE labeled liposomes showed green, and nuclei stained with Hoechst 33342 showed blue.

**Figure 3 molecules-24-03332-f003:**
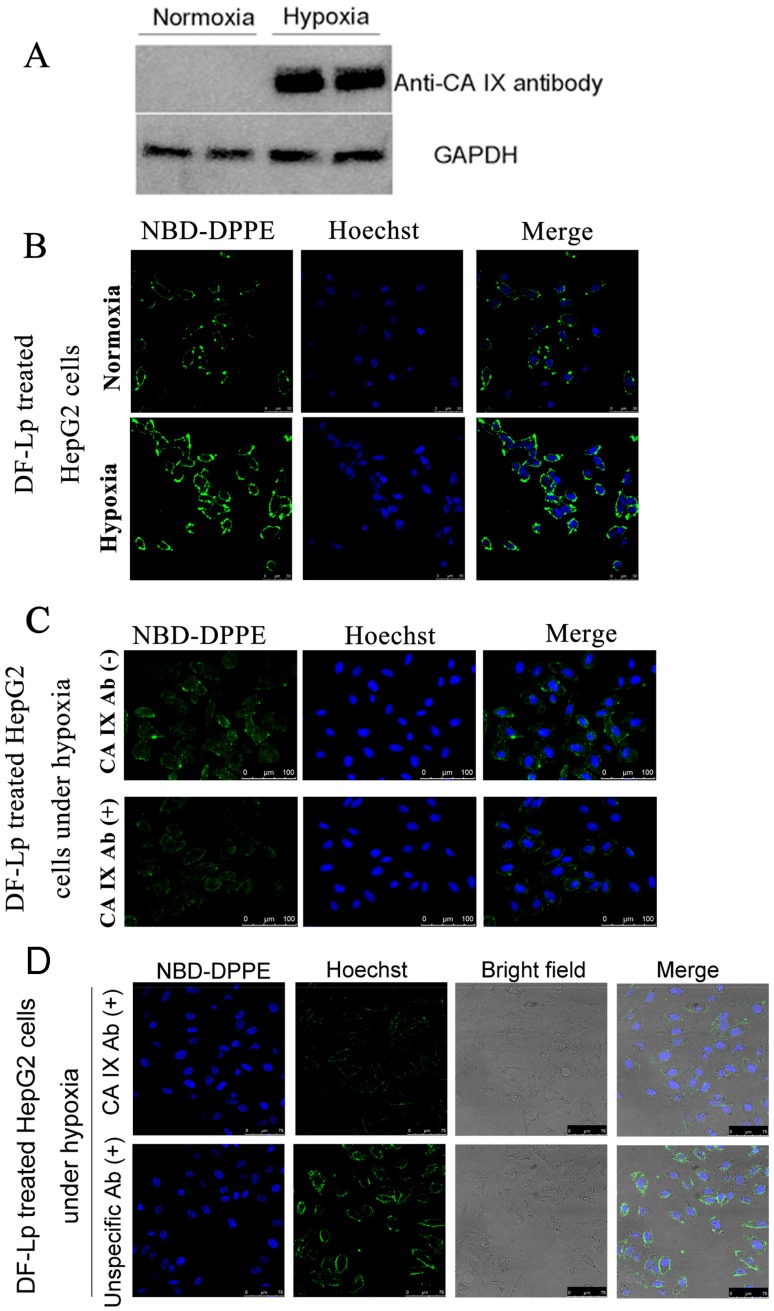
(**A**) Western blot analysis of CA IX expression in hypoxic and normoxic HepG2 cells. (**B**) Fluorescent images of CA IX positive HepG2 under hypoxic and CA IX negative HepG2 under normoxic conditions after treatment with DF-Lp formulation. Scale bar = 50 μm. (**C**) In vitro competition assay to detect targeting of CA IX by DF-Lp in hypoxic HepG2 cells. Cells were pre-incubated with (+) or without (−) anti-CA IX Ab for 15 min and then incubated with DF-Lp for an additional 30 min. Liposomes are shown by green fluorescence of NBD-DPPE by fluorescent microscopy. Scale bar = 100 μm. (**D**) In vitro competition assay to detect specific targeting of CA IX by DF-Lp in hypoxic HepG2 cells. Cells were pre-incubated with (+) anti-CA IX Ab or with (+) unspecific Ab for 15 min and then incubated with DF-Lp for additional 30 min. Liposomes are shown by green fluorescence of NBD-DPPE by fluorescent microscopy. Scale bar = 75 μm.

**Figure 4 molecules-24-03332-f004:**
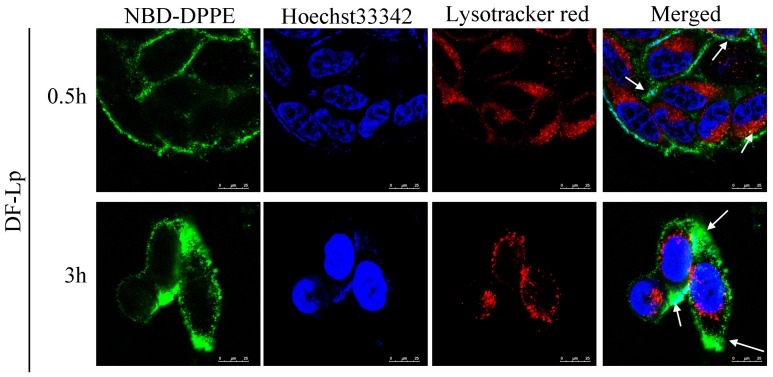
Colocalization of NBD-DPPE labeled dual-modified liposomes (DF-Lp, green) and endosomes/lysosomes (red). Nuclei were stained with Hoechst 33342 (blue). Scale bar = 25 μm.

**Figure 5 molecules-24-03332-f005:**
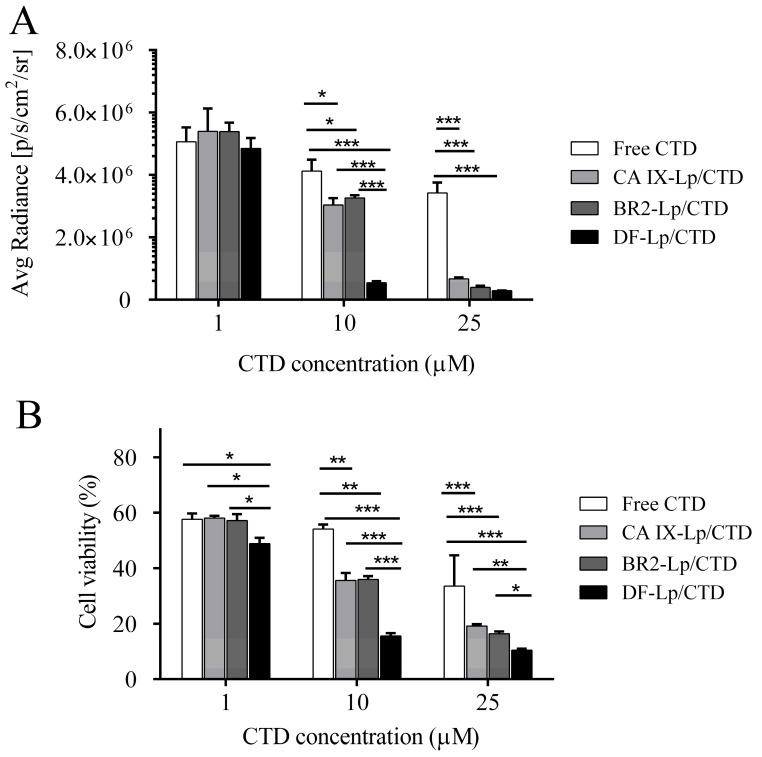
Cell viability assay of different liposomal formulations. HepG2-red-Fluc cells grown in black or transparent 96-well plate were treated with different CTD-liposomal formulations with CTD concentration at 1 µM, 10 µM, and 25 µM for 24 h. Then the bioluminescent intensity were tested by IVIS imaging systems (**A**) and the relative cell inhibitory effects were determined by MTT assay (**B**), respectively. * *p* < 0.05, ** *p* < 0.01, *** *p* < 0.001.

**Figure 6 molecules-24-03332-f006:**
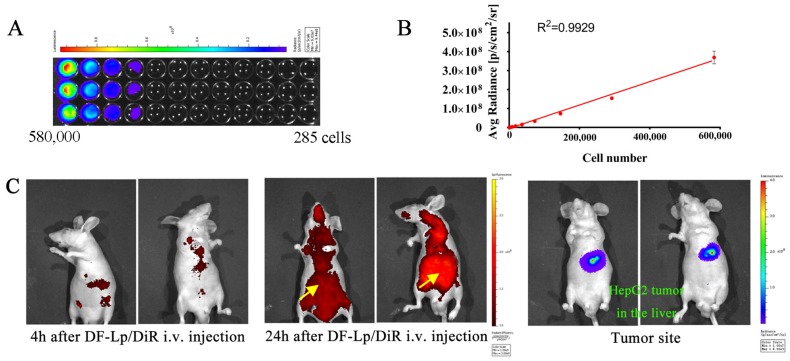
(**A**) In vitro imaging analysis of luciferase activity of HepG2-red-Fluc cells. (**B**) The photon flux plot as a function of increasing number of HepG2-red-Fluc cells obtained from IVIS each well (*n* = 3). (**C**) Representative in vivo imaging of the HCC orthotopic mice bearing luciferase-expressing tumors derived from HepG2-red-Fluc cells following i.v. injection of DF-Lp/DiR using the IVIS imaging system.

**Figure 7 molecules-24-03332-f007:**
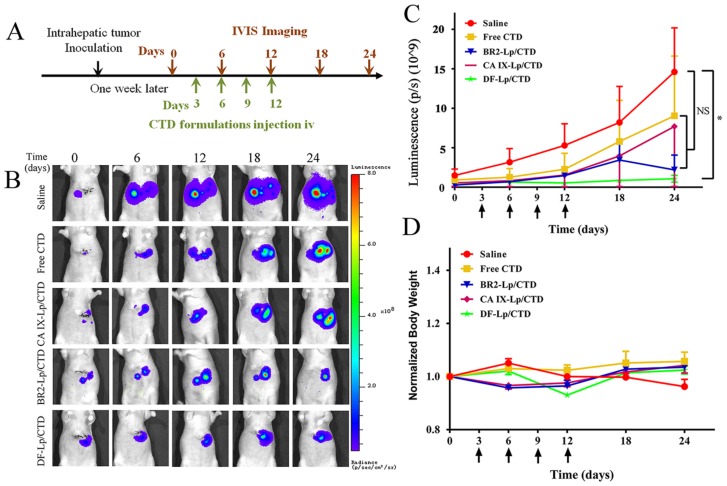
In vivo anticancer efficacy of DF-Lp/CTD on HepG2-red-Fluc orthotopic HCC tumors. (**A**) Scheme of HepG2-red-Fluc tumor inoculation and treatment at CTD dosage of 0.4 mg/kg. (**B**) Representative live IVIS images of HepG2-red-Fluc cell-bearing tumors with administration of D-luciferin with saline-treated mice served as a control. (**C**) Quantitative measures of luminescence (mean ± SEM) of the total flux of the tumors in the mice (*n* = 5). Arrows indicate drug administration time. * *p* < 0.05. (**D**) Bodyweight profiles of mice in different groups (*n* = 5, mean ± SEM).

**Table 1 molecules-24-03332-t001:** Characterization of different liposome formulations (mean ± SD).

Liposome Type	Liposome Component (Molar Ratio)	Particle Size (nm)	PDI
CA IX-Lp/CTD	SPC/DSPE-PEG/DSPE-PEG-Mal (95.8:3.8:0.4)	75.7 ± 1.5	0.135 ± 0.006
BR2-Lp/CTD	SPC/DSPE-PEG/DSPE-PEG-BR2 (96:2.0:2.0)	92.4 ± 1.2	0.261 ± 0.022
DF-Lp/CTD	SPC/DSPE-PEG/DSPE-PEG-BR2/DSPE-PEG-Mal (95.2:1.9:2.0:0.87)	98.3 ± 1.8	0.256 ± 0.003

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
