# Peer review of "Dual-Functional Liposomes with Carbonic Anhydrase IX Antibody and BR2 Peptide Modification Effectively Improve Intracellular Delivery of Cantharidin to Treat Orthotopic Hepatocellular Carcinoma Mice"

_molecules, 2019, doi:10.3390/molecules24183332_

Round 1
Reviewer 1 Report
This manuscript describes preparation and evaluation of liposomes with two targeting moieties – a peptide and an antibody to selectively effect HCC cells. The authors draw this as an improvement to their earlier single targeting moiety based drug delivery system. This seems to be the logical progression of the earlier study as the intracellular accumulation was not optimal. Overall the method description and data analyses seem appropriate. Nevertheless, there are several details missing in the manuscript, including appropriate study controls to make few conclusions. Please see suggestions and comments below.
1. The manuscript is written well and only contain minor editorial/grammatical errors. Suggest to include in vivo nature of this work in the title.
2. The active molecule as well as targeting moieties are not appropriately introduced. Please include a section on rationale for selection of each of these entities.
3. A major drawback of this study is the selectivity of the developed liposomes. It is understood that using CAIX under-expressing cells is one way to show that uptake will be somewhat specific in these cells. But the difference between hypoxia vs. normal conditions is not drastic as can be seen in supplemental information. Moreover, liposomes interact with other cells and as the authors rightly mentioned in the introduction, toxicity is an issue in cancer therapy due to non-specificity. Therefore, it is imperative that the effect of DF-liposomes should be shown in other cell lines (including phagocytic cells), where potential toxicities are excepted.
4. Have the authors optimized the overall ligand density on liposomes and also relative amounts of each targeting moiety?
5. Figure 1A is missing. BR2 conjugation diagram should be included.
6. GC MS is not described.
7. Although differences are seen between different liposomes species, how is it ensured that the amount of dye and the drug are matching between these entities? If there is more drug encapsulated or the dose is high, the tumor reduction will be high irrespective of the liposome type used.
8. How is drug release conducted, include procedure and method for analysis of drug. It can be in supplemental if the authors prefer that.
9. Please comment on how your DDS can improve resistance profile in this cancer model?
10. The efficacy is not drastically high especially until day 18, although 4 injections of liposomes have been completed by then. Please comment what is the reason for low efficacy?
11. Use abbreviations/acronyms consistently. Use appropriate symbol of “degree Celsius”.
Reviewer 2 Report
Manuscript ID: molecules-571093
Dual-functional liposomes with carbonic anhydrase IX antibody and BR2
peptide modification effectively improve intracellular delivery of
cantharidin to treat hepatocellular carcinoma
The paper by Xue Zhang et al. deals with the design of a novel liposome based nanoparticle directed to liver cancer. Such liposomes are loaded with a cytotoxic drug which is then delivered more efficiently that the free CTD to the tumor site. Nanoparticles have great potential as controllable drug delivery system. Notably, liposomes can be decorated with various targeting ligands and the authors used herein a liposome containing both a peptide and an antibody directed towards a membrane protein overexpressed in cancer cells. Such approach is interesting although not novel but is clearly of interest in order to improve our knowledge on the various possible targeting opportunities that unfortunately have been disappointing until now in the field of cancer. The main purpose of the authors is to compare liposomes decorated with a BR2 peptide with or without an antibody directed to carbonhydrase.
I have several major and minor concerns that the authors are encouraged to address in a revised version.
Major points:
More data on nanoparticles characterization and delivery of payload are needed.
Although particles are used with cells in complete medium for long time, the authors didn t report any experiment indicating the stability of the present nanoparticles and their capacity to retain the payload. Such information is clearly lacking.
In Figure 1, the presence of peptide and protein with the particle is revealed by bradford reaction. It is important to assess the presence of antibody by Western with secondary antibody and primary antibody for BR2 peptide. In addition, using acrylamide gel is unusual for nanaparticle characterization and interpretation of the observed shift in migration needs the addition of appropriate controls. The writting of legend on the gel is uncorrect.
In figure 2, the study of nanoparticles have been carried out without loading of CTD. In addition, the physico chemical characteristics of the nanoparticles have been carried when CTD is loaded. It is necessary to study the cellular delivery with short times of incubation of the various nanoparticles loaded with CTD and to report the physico chemical properties of both empty and loaded nanoparticles. This figure should be more easy to analyze if the authors add corresponding bright field images.
Figure 3C : the competition with anti-CA IX Ab should be made with unspecific antibody as a control in order to demonstrate that the competition with the nanopartcile is specific.
It is highly recommended to determine the role of antibody in increasing cellular delivery by performing an extensive study by cytometry. Without this important quantitative information, the interest of this study is significantly limited.
It is disappointing that the authors never quantify the intracellular delivery of CTD thanks to their new nanoparticle. If they cannot measure CTD inside cells by chromatography, they need to use a model molecule such as coumarin 6 as previously shown by the authors or any another drug. The entry of the vector into cells is clearly not sufficient to physically prove that the drug of interest is present at critical concentration into the target cancer cells. Page 7, line 273 it is this uncorrect to write the sentence : to assess the drug capacity …………….
In vivo :
Figure 6 : it is necessary to indicate by a circle or an arrow the increase of fluorescence at the tumor level induced by the liposome as depicted in the text (line 381) : » demonstrating the prolonged retention of liposomes in tumors ». The authors have overinterpreted their data because no biodistribution of the active drug has been reported. This point needs to be clearly indicated to avoid misinterpretation of the data.
Figure7 :
The authors need to indicate whether statistically a difference was observed between groups treated with BR2-Lp/CTD and groups treated with DF-Lp/CTD.
The conclusion from Figure 7 is overinterpreted by the authors: « Among all the CTD396 formulations, the DF-Lp/CTD exhibited the greatest and earliest therapeutic efficacy on tumor growth ». This sentence needs to be corrected as the primary tumor regression is not significantly different between BR2-Lp/CTD and DF-Lp/CTD even for days 18 and 24h. Moreover there is no difference for all the formulations for days 6 to 12. So it is highly suggested to the authors to clearly compare CTD alone with the different formulations and to conclude whether a difference is significative between the two main formulations DF-Lp/CTD and BR2-Lp/CTD.
A main problem in the analysis of the present data arise from important SD values. It should be of interest for the authors to solve such critical point by increasing the number of animals per group to 8 mice.
An important control is also lacking in this in vivo experiment in order to assess the possible anticancer effect of the empty nanoparticle or loaded with an inactive hydrophobic compound. It is well known that liposomes accumulate into liver rendering highly possible CTD independent anticancer effect.
Such experiments as indicated above are of great importance to increase the quality of the manucript.
Minor points :
Page 6 line 68 : carbonhydrase is not an antigen as written by the autors ; this enzyme is atransmembrane protein, not an antigen.
Line 72 : the sentence : « Considering…… » must be rewritten
Line 340 page 10 : »These results indicated the potent anticancer effects of this DF-Lp/CTD in vitro ». The term anticancer is wrong and must be removed.
Same correction to be made page 11 line 364
The authors have not indicated in the legend to Figure 7 the concentration of CTD loaded in nanoparticles used in vivo for IV injection
Round 2
Reviewer 1 Report
The authors have answered questions to my satisfaction. I recommend accepting this article for publication.
Author Response
Many thanks for the reviewer’s constructive suggestions.
Reviewer 2 Report
Manuscript ID: molecules-571093
Dual-functional liposomes with carbonic anhydrase IX antibody and BR2
peptide modification effectively improve intracellular delivery of
cantharidin to treat hepatocellular carcinoma
by Xue Zhang et al.
The authors have not addressed several major points previously raised; such points need to be considered for the clarity and quality of their work.
The premise of this revised version is that a new particle with potential increased capacity of cancer cell targeting should increase the anticancer activity of a cytotoxic drug. In response to the previous review, they indicate that such new particle doesn t need extensive characterization because it looks like to a particle reported in a previous work. This approximation cannot be accepted as both particles are significantly different.
Another point that the authors have to consider for publication is the characterization of particles that are loaded with the cytotoxic agent (DF-Lp/CTD), not of empty chemical vector. This point is also related to delivery experiment of the payload into cells that need to be verified with coumarine or other reporter drug in the present manuscript as the authors have indicated in their response that CDT detection is not possible.
1) Data on nanoparticles characterization and delivery of payload are needed as requested previously (major points 1,2, 4 ). I recommend also to add the bright field image in the manuscript, not in the supplementary material as it is important to observe the state of the transfected cells.
2) The major point 5 needs also to be verified as this experiment is easy to perform and help the authors to demonstrate the specific binding of the antibody and consequently the specificity of the particle which is a central point in this manuscript.
Author Response
Thank the reviewers for these precious comments and suggestions.
1) Data on nanoparticles characterization and delivery of payload are needed as requested previously (major points 1,2, 4 ). I recommend also to add the bright field image in the manuscript, not in the supplementary material as it is important to observe the state of the transfected cells.
We feel sorry to miss some major points. The appreance, particle size distribution, drug release, etc of CTD-liposomes are added to the supporting files already. Other properties are under investigation in our laboratory. Unfortunately, results are unavailable at this point.
And figure 2 has been replaced by the images including bright field images. Thank you for your comments.
We understand that delivery coumarin 6 may better reveal the delivery characterization. However, in the present study, we mainly focused on the delivery properties of our DF-Lp FOR CTD, and we think that NBD-DPPE labeled blank liposomes may not be optimal, but should be sufficient to draw a conclusion that delivery properties of our fabricated system.
2) The major point 5 needs also to be verified as this experiment is easy to perform and help the authors to demonstrate the specific binding of the antibody and consequently the specificity of the particle which is a central point in this manuscript.
We agree with the reviewer, and the figure 3 has been updated. The unspecific antibody pre-treated results has been added as well as in the results part.
Round 3
Reviewer 2 Report
The quality has been improved and the present manuscript by ZhangX. et al. is now acceptable for publication in Molecules.